# Does the "Returning Farmland to Forest Program" Drive Community-Level Changes in Landscape Patterns in China?

**Wenqing Li** [1,2] **, John Aloysius Zinda** [3] **and Zhiming Zhang** [1,*]

[1] School of Ecology and Environmental Sciences and Yunnan Key Laboratory for Plateau Mountain Ecology and Restoration of Degraded Environments, Yunnan University, Kunming 650091, China; liwenqing@mail.kib.ac.cn

[2] CAS Key Laboratory for Plant Diversity and Biogeography of East Asia, Kunming Institute of Botany, Chinese Academy of Sciences, Kunming 650201, China

[3] Department of Development Sociology, Cornell University, Ithaca, NY 14853, USA; jaz65@cornell.edu

[*] Correspondence: zhiming_zhang76@hotmail.com

**Abstract:** In China, the Returning Farmland to Forest Program (RFFP) has afforested large areas, transforming land and livelihoods. By impacting vegetation cover, it may also drive spatial pattern changes across landscapes. Most studies have focused on time series data as a means to determine the effectiveness of the program, but there is a paucity of community-level comparative studies. Twelve communities in Northwest Yunnan Province were selected to test whether the RFFP changed landscape patterns by testing the following hypotheses: with (or without) the RFFP, forest and shrubland fragmentations would decrease (or increase) and farmland fragmentation would increase (or decrease). Remote sensing images from 2000, 2010, and 2014 were used to compare the differences in landscape patterns. Survey data from 421 households were used to examine the socioeconomic and ecological factors that affect the differences in landscape fragmentation across communities. The results showed that landscape patterns and fragmentation metrics were not significantly different between communities with or without the RFFP, regardless of the class or landscape level. These communities showed consistent patterns of change in their fragmentation parameters between 2000 and 2014, with forest fragmentation decreasing and the fragmentation of farmland and the overall landscape increasing. The regression models suggest these changes were affected by the local natural conditions, socioeconomic patterns, policy implementation, and farmer livelihoods, with the proximity to market towns and elevation being significant factors. The RFFP alone did not directly drive the changes in landscape patterns for the considered region. For the new RFFP to effectively contribute to reducing fragmentation, managers of afforestation efforts should carefully consider livelihoods and biophysical factors that influence changes in landscape patterns.

**Keywords:** China; landscape pattern; reforestation; returning farmland to forest program; socio-ecological systems

## 1. Introduction

In 1999, to curb soil erosion and related forms of environmental degradation, the government of China launched the "Returning Farmland to Forests Program" (RFFP, also translated as the Sloping Land Conversion Program and the Grain for Green Program). The core of the project is to plant trees or grass on sloping and desertified lands withdrawn from grain production where soil erosion is serious or grain yield is low and unstable or important ecosystem services require protection [1]. By 2013, the government had invested ¥320 billion—involving 32 million households—in planting

trees over an area exceeding 29 million hectares [2,3]. Over the 18 years since its initiation, observers have attributed substantial ecological and social benefits to the RFFP, although concerns including biologically poor monoculture plantations, uneven survival rates, and unfairness in implementation have also arisen [4–6]. The three major goals of the policy are a stable supply of forest products, the improvement of rural livelihoods, and the protection and restoration of ecological systems. However, conflicts and contradictions can arise among these goals, and differences in the effectiveness of implementation are evident across locations and at various scales [7,8].

The spatial distribution of changes in forest cover at different scales varies, and considerable uncertainty exists regarding the effectiveness of the policy in increasing forest cover [9]. At the large scale, research shows that the RFFP has increased forest cover and changed the type of land use [10–17]. Simultaneously, the policy contributes to structural changes in rural economies and increases in household incomes [18–20]. However, small-scale studies suggest that ecosystem service programs and policies that encourage forest restoration have not been entirely successful and that changes in forest cover vary widely [21,22]. The direction of forest cover change is inconsistent, even decreasing in some areas [18,23,24]. In a separate study on the impact of the RFFP, Trac, et al. [25] found that it did not achieve the expected ecological benefits at the township level because of economic problems or inappropriate selection of tree species. Recent studies suggest that other factors that covary with RFFP implementation may account for much of the vegetation gains attributed to the program [26]. With differences in the effectiveness of the RFFP in promoting increased forest cover, the changes in landscape patterns are also likely to differ [27–29]. Therefore, further considerations should be given to the question of whether the RFFP drives community-level changes in landscape patterns in China.

The landscape is a perfect example of where the combined effects of society and nature become visible. As societies and nature are dynamic, change is an inherent characteristic of landscapes [30]. The drivers of changes in land use and the landscape pattern have been examined in many studies [28,30,31]. For instance, the effect of the RFFP on landscape patterns at a large scale revealed that forest fragmentation has decreased and the fragmentation of both farmland and ecosystem pattern has increased [32–34]. However, the RFFP affects landscape patterns differently at the medium and small scales [35]. This research shows that the implementation of the RFFP varies across regions [36]. Even within the same county, farmers in different towns and villages may receive different subsidies and show a varying willingness to participate. At a small scale, studies show that households who take part in the RFFP will reallocate farmland [35,37], but the evidence also shows that those who are not involved do the same [18,38]. These results suggest that other factors have a greater impact on farmland reallocation systems than RFFP, and RFFP may not directly affect these decisions [39]. In addition to the RFFP, socioeconomic, political, technological, ecological, and cultural factors may be primary drivers of landscape pattern change [36,40–43]. The study of Weng [44] demonstrated the impact of urbanization on landscape change by integrating the spatial and the temporal perspectives.

Most studies of RFFP focus on the time series analysis of large-scale landscape patterns, but the transect analysis and different patterns that exist at small scales are ignored. In this study, to evaluate the role of the RFFP in affecting the landscape pattern at the community level, 12 communities were selected in Weixi Lisu Autonomous County, Diqing Prefecture, with five communities that implemented the RFFP and seven communities that did not. The results of earlier studies in this region indicate that forest vegetation cover increased significantly from 2000 to 2014 [26,43]; however, no significant correlation was detected between the implementation of the RFFP and changes in vegetation cover at the community level. Therefore, the RFFP was not a direct cause of the changes in forest cover, although the combination of the RFFP and other factors may have played a role. Changes in forest cover inevitably cause changes in landscape patterns [9,45], and the regional landscape pattern is also likely impacted by the requirement of relevant government departments for continuous land conversion [25,46,47]. It is anticipated that widespread farmland retirement and the restoration of vegetation under RFFP would reduce forest fragmentation but increase farmland fragmentation. Based on these expectations, the landscape patterns between communities with and without RFFP should

be different. Thus, this study addressed the following questions: Is there a significant difference in landscape patterns and fragmentation between communities that did and did not implement the RFFP in 2000, 2010, and 2014? Is there a significant difference in landscape pattern changes between the communities that did and did not implement the RFFP from 2000 to 2014? Are the changes in land use cover and landscape pattern caused by the implementation of the RFFP? Expanding on our previous work, which was limited to the change in forest cover, this study analyzes the impact of the RFFP on farmland, shrubland, forests, and the entire landscape pattern at the community level and explores the driving forces of landscape pattern change, combining environmental and socioeconomic data. The results of this study can provide guidance for the smooth development of forest restoration policies succeeding the RFFP and also serve as a basis for evaluating the impacts of RFFP relative to policy goals. More specifically, the study can provide a scientific basis for the planning and utilization of land resources in Weixi County and a reference point for realizing sustainable development in this region.

## 2. Materials & Methods

### 2.1. Study Area

Weixi County (98°54′–99°34′ E, 26°53′–28°02′ N) is a key area for implementing the RFFP and the Natural Forest Protection Program (NFPP). The area has highly rugged terrain, with a maximum elevation of 4880 m. The Lancang River flows through the county from north to south at an elevation from 1380 to 1800 m. The county has a temperate monsoon plateau–mountain climate. Although the county is rich in biological and cultural diversity, it is also designated a national-level poverty county. Since the implementation of the RFFP in 2001, the protection of biodiversity and the livelihoods of farmers have undergone significant changes, reflecting the prevailing situation of rural areas in Southwest China. Weixi County has jurisdiction over seven townships and three towns. Within a given town or township, some communities participated in the RFFP and some did not. This variation in participation, along with varying outcomes in implementing communities, make it a strong site for examining varying landscape pattern impacts of the program. In this study, 12 communities (each a natural village, a basic administrative unit) in two adjacent townships of Weixi County were selected, including five communities that participated in the RFFP and seven that did not. To control for the influence of sunlight and precipitation on forest growth, communities were selected that were located east or southeast of the west bank of the basin. In each of two townships, six communities were selected: two located at the riverside, two with one community separating them from the river, and two at high elevation, with two or more communities separating them from river (Figure 1). In the presentation below, all names of places and persons are pseudonyms.

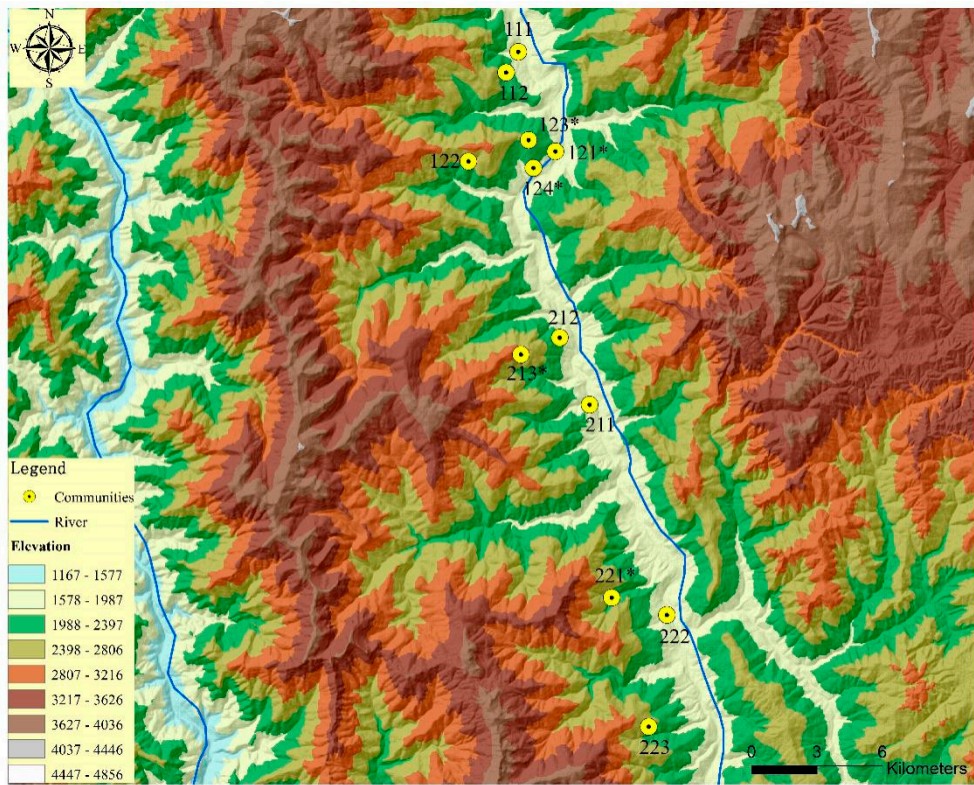

**Figure 1.** Study location. * indicates communities that implemented the Returning Farmland to Forest Program (RFFP).

## 2.2. Remote Sensing Data

Three remotely sensed images of Weixi County were acquired: Landsat Enhanced Thematic Mapper Plus (ETM+) data for 2000 with a spatial resolution of 30 m, Advanced Land Observation Satellite (ALOS) data for 2010 with a spatial resolution of 10 m, and Gaofen-1 data for 2014 with a spatial resolution of 8 m (Table 1). Each image was acquired between November and January. The 2000 image data reflect the landscape pattern before the implementation of the RFFP. The 2010 image represents the pattern during the implementation of the RFFP. The year 2010 also marks a period during which farmers began to plant walnuts and other cash crops on a large scale. The 2014 image represents the pattern of the current landscape. The data for these three periods were conducive to testing the impact of the RFFP on land use and landscape patterns in Weixi County.

**Table 1.** Data source. ETM+: Landsat Enhanced Thematic Mapper Plus; ALOS: Advanced Land Observation Satellite.

| Data Acquisition Time | Sensor | Spatial Resolution of Multi-Spectral Band (m) |
|---|---|---|
| 25 December 2000 | Landsat ETM+ | 30 |
| 4 January 2010 | ALOS | 10 |
| 8 November 2014 | Gaofen-1 | 8 |

To resolve the inconsistent spatial resolution of the remote sensing sources, resampling was performed using the 30 m resolution of the images in 2000 to ensure consistent resolution across the three times periods. To correct for geometric and radiation distortions, geometric correction and radiation correction were preprocessed, respectively [48]. The three images were classified using a combination of object-oriented interpretation, visual interpretation, and an artificial neural network feed-forward back-propagation algorithm [49,50], with the specific methods detailed in Zhang, et al. [51]. Based on

the first-level classification system of land use status [52] and field investigations in Weixi County, four land use categories were identified: agricultural land, forest, water, and other. Because the study region is the Hengduan Mountain range in the northwest of Yunnan Province, the remote sensing images had areas with mountain shadow, and this type of image was classified separately. Therefore, six categories were in the secondary classification: agricultural land, forest, shrubland, water, snow, and cast shadow (Table 2). The confusion matrix method was used to evaluate the accuracy of image classification, and the overall accuracy and Kappa coefficients were used as the indicators to detect the classification results [53,54] (Table S1). In 2014, global positioning system (GPS) point sampling in a field survey was used, with the categories forest (200), shrubland (40), agricultural land (80), and river (10). In other years, forest, shrubland, and agricultural land were sampled at 30 points, and the categories of other, water, snow, and shadow were sampled at 15 points. The overall accuracies of the classification results in 2000, 2010, and 2014 were 87.33, 89.33, and 89.33%, with the Kappa coefficients of 84.84, 87.24, and 87.27%, respectively. With an accuracy higher than 80% in each year, the classification results of the three phases were highly reliable and met the requirements for further analysis [55].

**Table 2.** Land cover classes.

| Class | Description |
| --- | --- |
| Forest | Fir and spruce forest; pine forest; mixed forest |
| Shrubland | Low density forest and tall shrubs; dwarf shrubs and meadow |
| Agricultural land | Agricultural land |
| Snow | Snow |
| Water | Water |
| Cast shadow | Cast shadow |

## 2.3. Landscape Pattern

The landscape feature metrics can reflect changes in the landscape pattern, represent the composition and configuration of the landscape structure, and present the spatial characteristics of different patterns at different levels [56,57]. Generally, the characteristics of the landscape pattern are analyzed at two levels: class and landscape. The variation in landscape fragmentation includes the changes in landscape patch size, quantity, shape, and consistency in time and space [58]. The selection of landscape metrics must consider their ecological significance, the correlations between indices, their sensitivity to spatial and temporal differentiation of a landscape pattern, and their sensitivity to remote sensing data resolution [59]. Based on the above considerations and the specific situation of Weixi County, representative metrics were selected from the two levels of basic landscape features and fragmentation [60,61]. At the class level, five indices were selected: tge number of patches (NP), mean patch area (MPS), fractal dimension (FRAC), connectance (CONNECT), and class fragmentation index (FN). At the landscape level, six indices were selected: number of patches (NP), mean patch area (MPS), fractal dimension (FRAC), connectance (CONNECT), Shannon diversity index (SHDI), and landscape fragmentation index (LFI) (Table 3). The ecological significance of these selected landscape indices is discussed in Xing and Shen [62] and Gu, et al. [63]. The landscape index analysis software Fragstats v4.2 [64] was used to calculate the landscape indices.

**Table 3.** Main landscape indices [65].

| Landscape Metrics | Formula | Basis for Selection |
|---|---|---|
| Mean patch area | $\text{MPS} = \frac{\sum_{j=1}^{n} a_{ij}}{n_i} \left(\frac{1}{10,000}\right)$ | The most basic spatial feature of landscape patterns and reflects the degree of landscape fragmentation [66]. |
| Number of patches | $\text{NP} = n_i$ | Most directly reflects the separation and fragmentation of a landscape or class [67]. |
| Fractal dimension index | $FRAC = \frac{2\ln(0.25 p_{ij})}{\ln a_{ij}}$ | Primarily used to measure the complexity of patch shape and also reflects the fragmentation of a landscape. A value close to two indicates that a patch shape is more complex [68]. |
| Connectance | $CONNECT = \frac{\sum_{j \neq k}^{n} c_{ijk}}{\frac{n_i (n_i-1)}{2}}$ | Used to describe the connectivity of the same type of patch in a landscape [69]. |
| Fragmentation index | $FN = \frac{\text{NP}}{\text{CA}} \quad LFI = \frac{\text{NP}}{\text{TA}}$ | FN is the fragmentation index of a class, and LFI is the landscape fragmentation index of an entire region. They reflect the degree of landscape segmentation. The greater the value is, the higher the landscape fragmentation [70,71]. |
| Shannon's diversity index | $\text{SHDI} = -\sum_{i=1}^{n} p_i \ln p_i$ | Reflects the richness and evenness of landscape patches and may also reflect the heterogeneity of a landscape [68]. |

In formulas, $i$ is the $i$th land cover type; $j$ is the $j$th patch of the $i$th land cover type; $a_{ij}$ is the $j$th patch area (ha) of the $i$th land cover type; $n_i$ is the patch number of the $i$th land cover type; $p_{ij}$ is the $j$th patch perimeter (m) of the $i$th land cover type; $c_{ijk}$ is the joining between patch $j$ and $k$ (0 = unjoined, 1 = joined) of the corresponding patch type (i), based on a user-specified threshold distance; $n_i$ is the number of patches in the landscape of the corresponding patch type (class); CA is the class area; TA is the total area; $p_i$ is the proportion of the landscape occupied by patch type (class) $i$.

## 2.4. Interview and Survey Data

The socioeconomic data were collected by household surveys, focus group discussions, and in-depth interviews with officials and members of selected households. Pilot surveys were conducted in January and June 2014. In July and August 2014, the authors and 11 professionally trained enumerators conducted research on each community.

1. *Household Surveys*

The household survey targeted all households present in each community, with 15–100 respondents in each research site, resulting in 419 valid questionnaires—a response rate of 80%. Interviews were conducted with adult members of each household, aged 18–65. The questionnaire covered demographic attributes of household members (gender, age, education level, labor activities) as well as livelihoods and land use activities (labor allocation, land use, and income related to crop cultivation, livestock husbandry, forest product harvesting, and off-farm work), and forest use and RFFP participation. Interviews were conducted at the home of the interviewee or at a location chosen by the interviewee.

2. *Focus Group Discussions and Intensive Interviews*

To obtain in-depth information on community participation in the RFFP and on resource utilization, within each community, six households were selected to participate in a focus group discussion [72]. In consultation with local village officials, participants were chosen who varied in age, gender, and wealth. Discussions focused on each community's history of changing livelihoods, forest management, and community affairs. Each discussion was recorded in a notebook and on a digital recorder with the informed consent of interviewees and lasted between 45 and 90 min. During a follow-up visit in autumn 2014, semi-structured interviews were conducted with a random sample of

households in each community, stratified by income terciles based on survey results. These interviews examined household decisions on land use and labor allocation, as well as community history and institutions. A total of 71 household interviews were conducted. Twenty-four semi-structured interviews were also conducted with the officials in the community, administrative village, township, and county agencies. The interviews covered the individual's responsibilities and experiences as an official, as well as the local implementation of policies and other community affairs. All interviews were conducted in Mandarin.

*2.5. Statistical Analyses*

Questionnaire data and interview records were imported into NVivo v10 software for integration and management. These data were coded and classified by themes, and differences across communities were compared. Questionnaire data were entered into a spreadsheet for processing. Household data were aggregated into community-level measures of crop cultivation, livestock husbandry, RFFP participation, and non-farm labor allocation. Landscape structure and fragmentation data from the remote sensing analysis were incorporated into the dataset. Variables were constructed measuring change in landscape metrics and fragmentation metrics for agricultural land, shrubland, and forest within each community. The results of tests for normality, homoscedasticity, skewness (=0), kurtosis (≈0), and multicollinearity (<0.8) were all within acceptable ranges. The 12 communities in the study area were divided into two groups: RFFP and non-RFFP. Student's *t*-tests [73] were used at class and landscape levels to evaluate whether the landscape metrics of the two groups were significantly different. To explore the driving forces of landscape fragmentation pattern changes in the study area, ordinary least squares regression models (OLS) [74,75] were used with forest, shrubland, agricultural land, the entire landscape, and environmental variables (environmental, policy, and social economy indicators) in 2010–2014 (Table 4). Before modeling, to avoid the multicollinearity of variables, which leads to over-fitted models, with the highly correlated variables affecting model accuracy, we used the stepwise regression method. The criteria for variables entering the model was the probability of $F \leq 0.05$, and the probability of $F \geq 0.10$ was removed. All statistical analyses were performed in R v3.5.1 (http://www.r-project.org/). The variables used in the models are described in Table 4.

**Table 4.** Variable descriptions.

| Variable | Description |
| --- | --- |
| Change in forest fragmentation in 2010–2014 | Rate of change in forest fragmentation of a community |
| Change in shrubland fragmentation in 2010–2014 | Rate of change in shrubland fragmentation of a community |
| Change in farmland fragmentation in 2010–2014 | Rate of change in farmland fragmentation of a community |
| Change in landscape fragmentation in 2010–2014 | Rate of change in landscape fragmentation of a community |
| Mean elevation | Mean elevation of all pixels within adjusted community boundaries |
| Household density | Number of households in 2014 divided by the community area in square kilometers |
| RFFP implementation | Binary: 1 = implemented RFFP, 0 = did not implement RFFP |
| Off-farm proportion | Proportion of surveyed households reporting off-farm labor in 2013 |
| Nearness to township | Binary: 1 = near, 0 = far |
| Solar water heater proportion | Proportion of surveyed households reporting installing a solar water heater by 2013 |
| Walnut area | Area of walnut tree plantation in a community: sum of household responses adjusted for nonresponse rate |
| Cropland retirement proportion | Proportion of households reporting cropland retirement not due to RFFP |
| Planted trees proportion | Average ratio of recent tree planting area to household cropland area |

## 3. Results

### 3.1. Changes in Land Cover and Landscape Patterns

Land cover and landscape patterns changed appreciably in all 12 communities between 2000 and 2014 (Figures 2 and 3). Forest, shrubland, and agricultural land occupied the largest area in the entire landscape and played leading roles in the changes in land cover and landscape pattern. From 2000 to 2010, whether forest and shrubland coverage increased or decreased across the 12 communities varied, whereas agricultural land area decreased in all communities. Between 2010 and 2014, except for one community that implemented the RFFP (123*), the forest cover of the 11 other communities increased. Shrubland coverage decreased in 75% of the communities. In addition to the two communities that were without RFFP (112 and 223), the agricultural area of the 10 other communities decreased in this period (Figures 2 and 3).

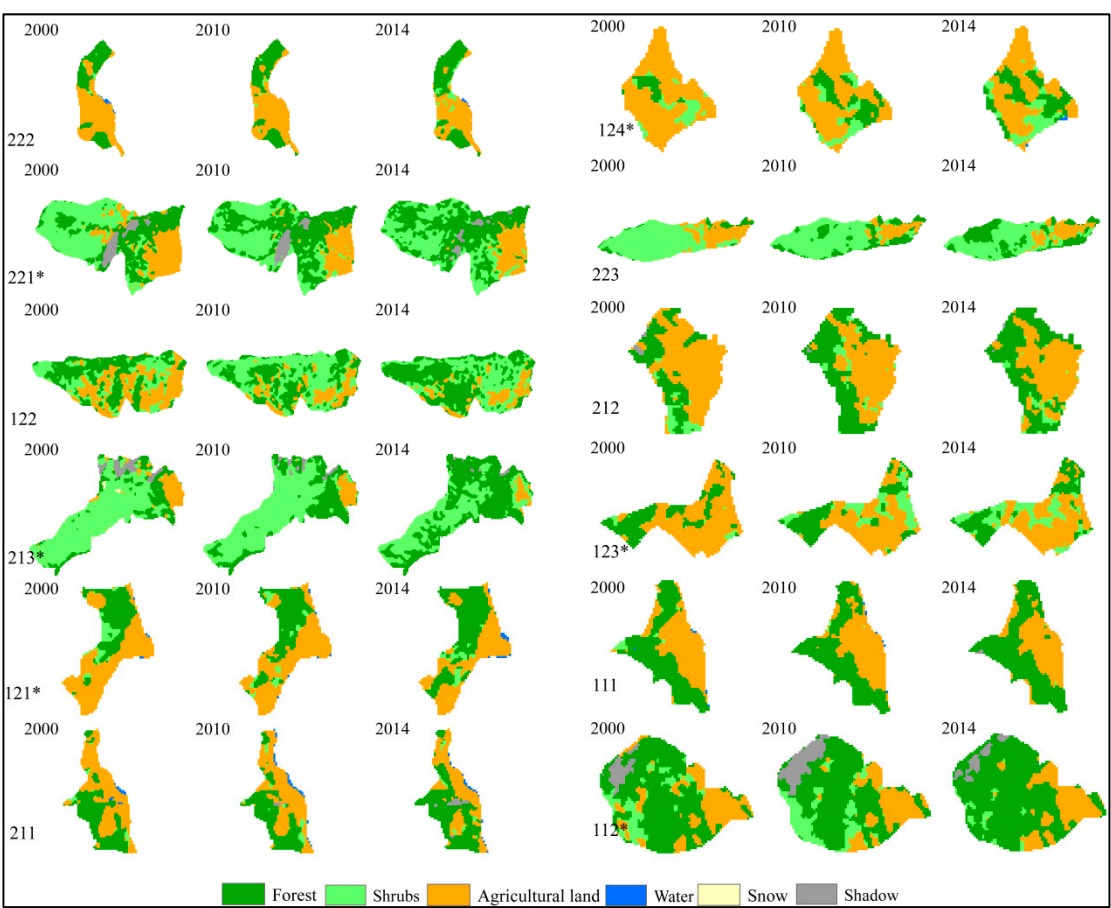

**Figure 2.** The distribution and composition of land cover for each community during three periods. * indicates communities that implemented RFFP. The map was plotted in ArcGIS v10.2.

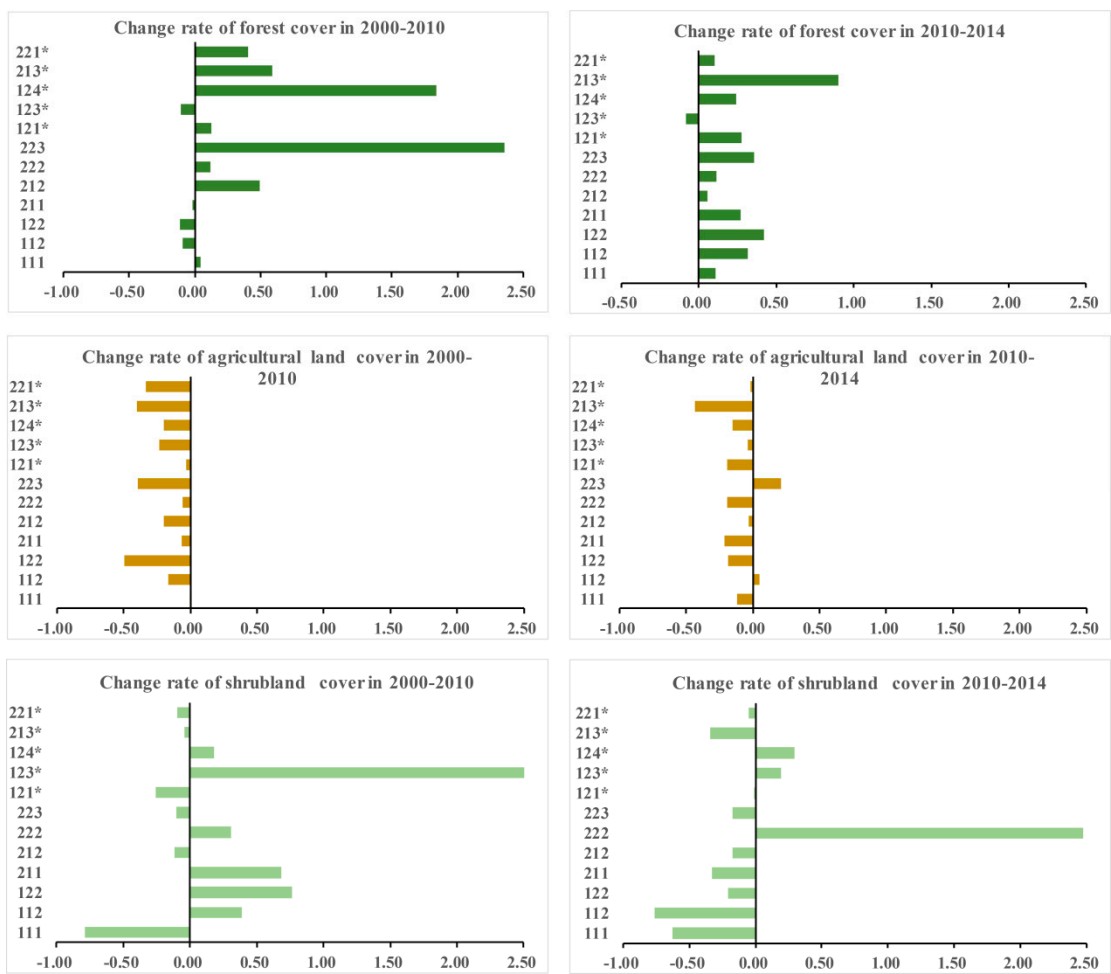

**Figure 3.** Change rate of forest, shrubs and agricultural land cover for each community in 2000–2010 and 2010–2014. * indicates communities that implemented RFFP.

## 3.2. Comparison of Landscape Metrics between RFFP and Non-RFFP Communities

The differences in landscape patterns and fragmentation between RFFP and non-RFFP communities were examined in 2000, 2010, 2014, and for 2000–2010, 2010–2014, and 2000–2014. For these time periods, the likelihood that the RFFP could produce changes in landscape patterns immediately after implementation was assessed. The different time periods also allowed sufficient time to assess possibly delayed effects on longer-term changes in plant growth and land use cover.

### 3.2.1. Landscape Metric Comparisons at the Landscape Level

During 2000–2010 and 2010–2014, the rate of landscape index change at the landscape level varied and showed increases and decreases in both RFFP and non-RFFP communities (Figure 4). In further analysis, no significant differences in the NP, LFI, MPS, FRAC, CONNECT, and SHDI indices were detected between the two groups in 2000, 2010, and 2014 ($p > 0.05$) (Table S2). Similarly, no significant differences were detected in the change rate of the indices during 2000–2010, 2010–2014, and 2000–2014 ($p > 0.05$) (Table 5). Thus, with the same dynamic changes occurring without significant differences in the landscape patterns of RFFP and non-RFFP communities, these results indicated that the RFFP might not be the primary driving force for the change in overall landscape patterns.

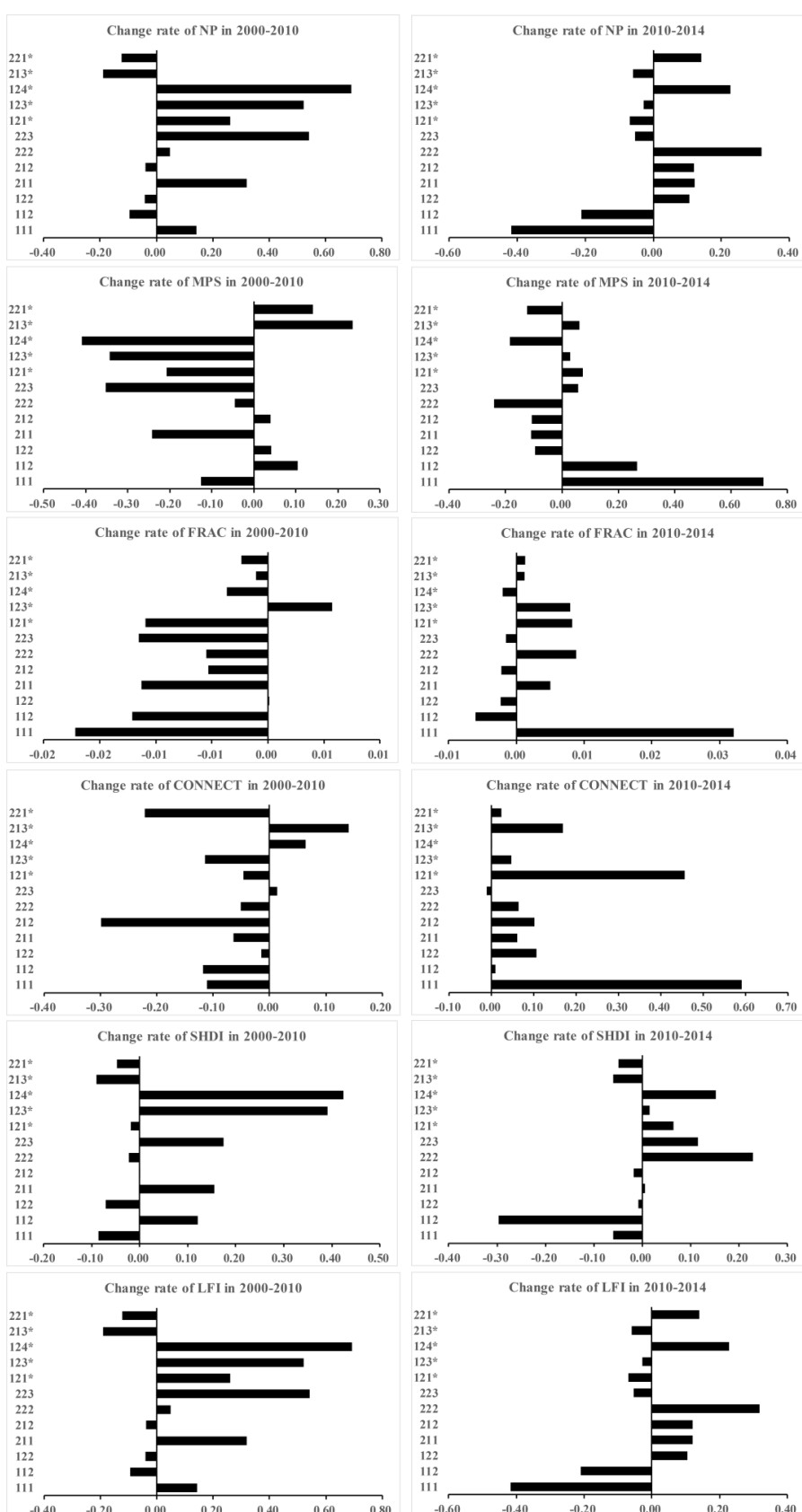

**Figure 4.** Change rate of landscape indices for each community in 2000–2010 and 2010–2014. * indicates communities that implemented RFFP. Refer to Table 3 for the abbreviations for landscape metrics.

**Table 5.** Comparison of landscape indices between RFFP and non-RFFP communities.

| | RFFP | | Non-RFFP | | | |
|---|---|---|---|---|---|---|
| | Mean | Standard Deviation | Mean | Standard Deviation | *t* | *p* |
| **2000** | | | | | | |
| NP | 42.200 | 32.453 | 36.857 | 28.328 | −0.304 | 0.768 |
| MPS | 5.907 | 1.253 | 5.888 | 1.730 | −0.020 | 0.984 |
| LFI | 0.175 | 0.034 | 0.182 | 1.253 | 0.275 | 0.789 |
| FRAC | 1.056 | 0.007 | 1.059 | 0.011 | 0.635 | 0.540 |
| CONNECT | 42.529 | 19.462 | 45.761 | 16.129 | 0.315 | 0.759 |
| SHDI | 0.903 | 0.235 | 0.901 | 0.130 | −0.020 | 0.985 |
| **2010** | | | | | | |
| NP | 43.000 | 22.305 | 39.143 | 25.452 | −0.272 | 0.791 |
| MPS | 5.399 | 2.674 | 5.256 | 1.172 | −0.127 | 0.901 |
| LFI | 0.221 | 0.091 | 0.200 | 2.674 | −0.516 | 0.617 |
| FRAC | 1.053 | 0.007 | 1.050 | 0.012 | −0.567 | 0.583 |
| CONNECT | 41.633 | 21.492 | 40.384 | 11.112 | −0.133 | 0.897 |
| SHDI | 0.981 | 0.118 | 0.934 | 0.149 | −0.600 | 0.575 |
| **2014** | | | | | | |
| NP | 45.000 | 26.048 | 39.714 | 29.719 | −0.319 | 0.756 |
| MPS | 5.217 | 2.523 | 5.626 | 2.037 | 0.311 | 0.762 |
| LFI | 0.231 | 0.103 | 0.201 | 0.077 | −0.579 | 0.575 |
| FRAC | 1.057 | 0.004 | 1.055 | 0.007 | −0.617 | 0.551 |
| CONNECT | 47.519 | 24.412 | 45.744 | 14.964 | −0.157 | 0.878 |
| SHDI | 1.000 | 0.092 | 0.915 | 0.113 | −1.386 | 0.196 |

Refer to Table 3 for the abbreviations for landscape metrics.

### 3.2.2. Landscape Metric Comparisons at the Class Level

Forest metrics: During 2000–2010, the rate of change in forest landscape indices varied across the 12 communities (Figure 5). From 2010 to 2014, 75% of community forests showed a decrease in the NP and LFI, an increase in the MPS and CONNECT, and a very slight change in the FRAC. However, no significant differences were detected in forest metrics or their rate of change between RFFP and non-RFFP communities in 2000–2014 (Student's *t*-test, *p* > 0.05) (Table 6 and Table S3). Based on the absence of differences, the RFFP had no significant effect on the change in forest landscape patterns. Similarly, the dynamic changes in forest patterns showed no differences between RFFP and non-RFFP communities during 2000–2010 and 2010–2014. After 2010, the forest fragmentation was inhibited in both groups. Therefore, irrespective of whether the RFFP was implemented, the same dynamic changes were observed in the forest landscape patterns, indicating that the RFFP was not the primary driving force promoting the benign evolution of the forest landscape.

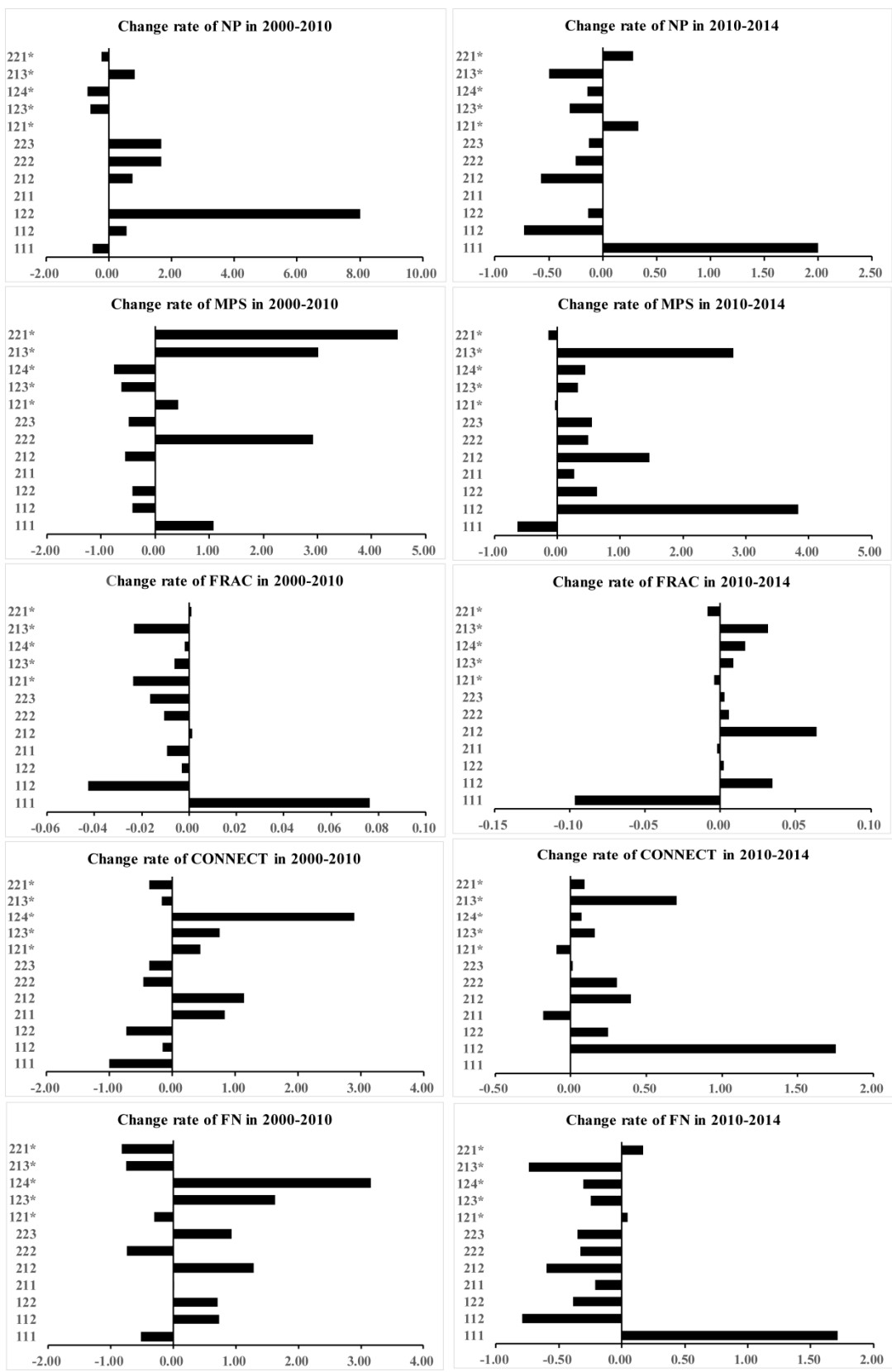

**Figure 5.** Change rate of forest landscape indices for each community in 2000–2010 and 2010–2014. * indicates communities that implemented RFFP. Refer to Table 3 for the abbreviations for landscape metrics.

**Table 6.** Comparison of forest landscape indices between RFFP and non-RFFP communities.

| | RFFP | | Non-RFFP | | | |
|---|---|---|---|---|---|---|
| | Mean | Standard Deviation | Mean | Standard Deviation | *t* | *p* |
| 2000 | | | | | | |
| NP | 13.200 | 10.686 | 9.710 | 9.878 | −0.583 | 0.573 |
| MPS | 4.939 | 3.160 | 12.087 | 10.514 | 1.456 | 0.176 |
| FN | 0.286 | 0.178 | 0.210 | 0.272 | −0.546 | 0.597 |
| FRAC | 1.065 | 0.008 | 1.079 | 0.022 | 1.409 | 0.189 |
| CONNECT | 43.506 | 20.279 | 45.838 | 27.219 | 0.161 | 0.875 |
| 2010 | | | | | | |
| NP | 13.000 | 9.721 | 15.143 | 14.276 | 0.289 | 0.778 |
| MPS | 18.232 | 27.263 | 6.209 | 2.911 | −1.181 | 0.265 |
| FN | 0.225 | 0.206 | 0.202 | 0.117 | −0.247 | 0.810 |
| FRAC | 1.078 | 0.048 | 1.061 | 0.010 | −0.948 | 0.365 |
| CONNECT | 33.403 | 32.717 | 47.772 | 26.878 | 0.836 | 0.423 |
| 2014 | | | | | | |
| NP | 12.200 | 8.468 | 11.571 | 13.685 | −0.090 | 0.930 |
| MPS | 8.705 | 6.688 | 17.601 | 17.289 | 1.082 | 0.305 |
| FN | 0.182 | 0.119 | 0.112 | 0.098 | −1.113 | 0.292 |
| FRAC | 1.069 | 0.019 | 1.074 | 0.033 | 0.319 | 0.757 |
| CONNECT | 51.823 | 29.572 | 65.069 | 35.039 | 0.686 | 0.508 |

Refer to Table 3 for the abbreviations for landscape metrics.

Shrubland metrics: In 2000–2010 and 2010–2014, the rate of change in shrubland landscape indices varied across the 12 communities and did not show consistent change (Figure 6). No significant differences in the NP, FN, MPS, FRAC, and CONNECT indices were observed between RFFP and non-RFFP communities in 2000, 2010, and 2014 ($p > 0.05$) (Table 7). The change rates of these indices were also not significantly different between the two groups in 2000–2010 and 2010–2014 ($p > 0.05$) (Table S4). These results showed that the change in the shrubland landscape pattern was consistent between RFFP and non-RFFP communities in the periods 2000–2010 and 2010–2014. Thus, the RFFP may not be the direct cause of the change in the shrubland landscape pattern because its implementation did not significantly affect the shrubland landscape pattern or its dynamic change.

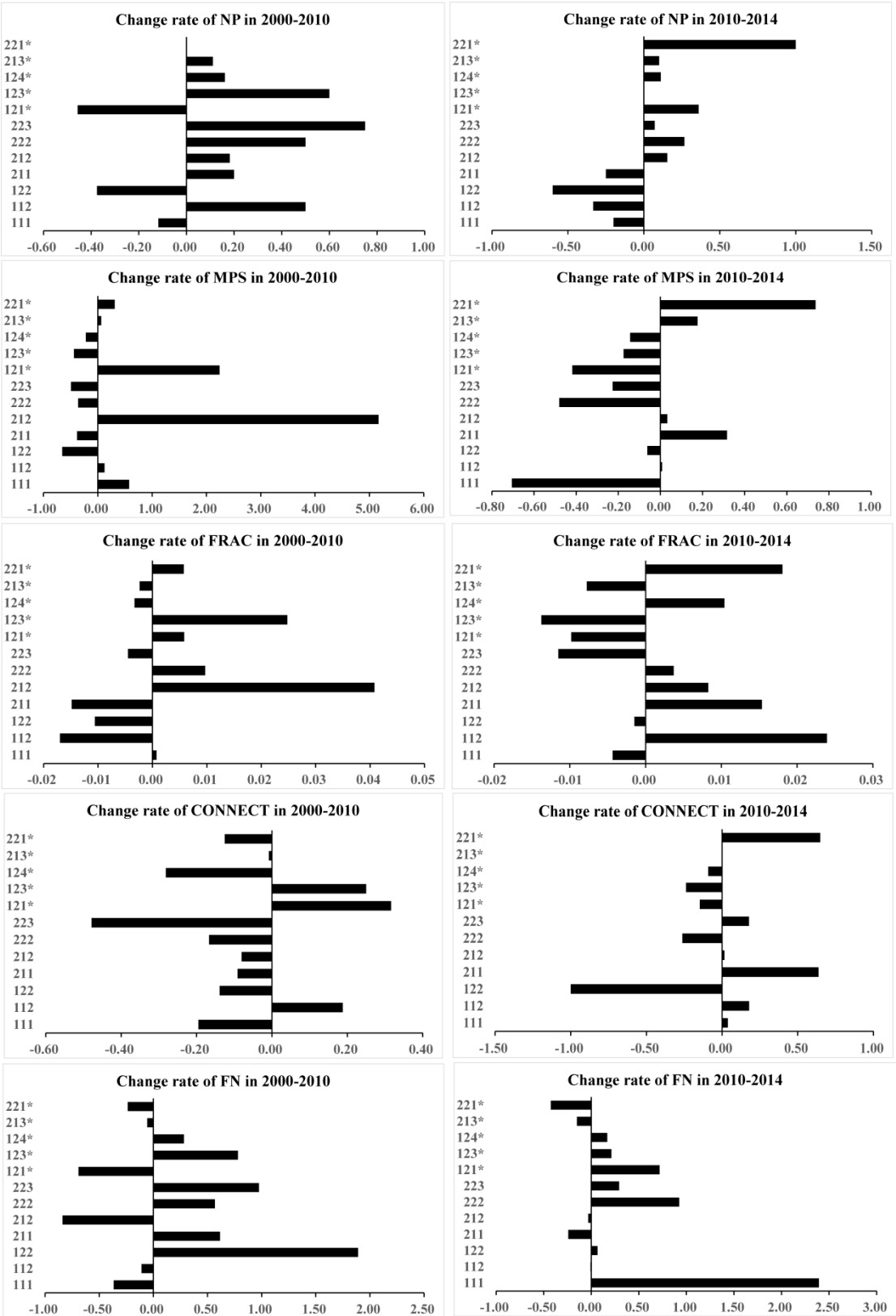

**Figure 6.** Change rate of shrubs landscape indices for each community in 2000–2010 and 2010–2014. * indicates communities that implemented RFFP. Refer to Table 3 for the abbreviations for landscape metrics.

**Table 7.** Comparison of shrubland landscape indices between RFFP and non-RFFP communities.

| | RFFP | | Non-RFFP | | | |
|---|---|---|---|---|---|---|
| | Mean | Standard Deviation | Mean | Standard Deviation | $t$ | $p$ |
| 2000 | | | | | | |
| NP | 14.200 | 9.418 | 14.143 | 14.554 | −0.008 | 0.994 |
| MPS | 7.406 | 10.615 | 5.384 | 11.181 | −0.315 | 0.759 |
| FN | 1.077 | 1.280 | 1.355 | 1.094 | 0.405 | 0.694 |
| FRAC | 1.045 | 0.020 | 1.044 | 0.017 | −0.104 | 0.919 |
| CONNECT | 44.532 | 16.165 | 47.934 | 20.241 | 0.310 | 0.763 |
| 2010 | | | | | | |
| NP | 17.200 | 10.663 | 12.286 | 6.726 | −0.985 | 0.348 |
| MPS | 5.425 | 6.535 | 4.607 | 6.782 | −0.209 | 0.839 |
| FN | 0.669 | 0.673 | 1.972 | 2.385 | 1.173 | 0.268 |
| FRAC | 1.051 | 0.009 | 1.045 | 0.026 | −0.505 | 0.625 |
| CONNECT | 40.177 | 18.580 | 43.156 | 15.617 | 0.302 | 0.769 |
| 2014 | | | | | | |
| NP | 18.800 | 12.458 | 13.286 | 10.144 | −0.846 | 0.417 |
| MPS | 3.758 | 3.400 | 3.309 | 5.287 | −0.166 | 0.872 |
| FN | 0.572 | 0.487 | 2.156 | 2.504 | 1.377 | 0.199 |
| FRAC | 1.058 | 0.009 | 1.045 | 0.020 | −1.295 | 0.225 |
| CONNECT | 43.660 | 24.770 | 39.865 | 21.277 | −0.285 | 0.781 |

Refer to Table 3 for the abbreviations for landscape metrics.

Agricultural land metrics: In 2000–2014, the rate of change in the various landscape indices of agricultural land varied across the 12 communities. From 2010 to 2014, 75% of community agricultural lands showed an increase in the NP and FN and a decrease in the MPS and CONNECT, and the remaining 25% of the communities had different change rates (Figure 7). For each agricultural land landscape index, no significant differences were observed between RFFP and non-RFFP communities in 2000, 2010, and 2014 ($p > 0.05$) (Table 8). During 2000–2010, 2010–2014, and 2000–2014, no significant differences were observed in the change rates of the agricultural land landscape indices ($p > 0.05$) (Table S5). These results showed that the same dynamic changes occurred in the agricultural landscapes of RFFP and non-RFFP communities, and the implementation of the RFFP was therefore not the key factor affecting the change in landscape patterns of agricultural land.

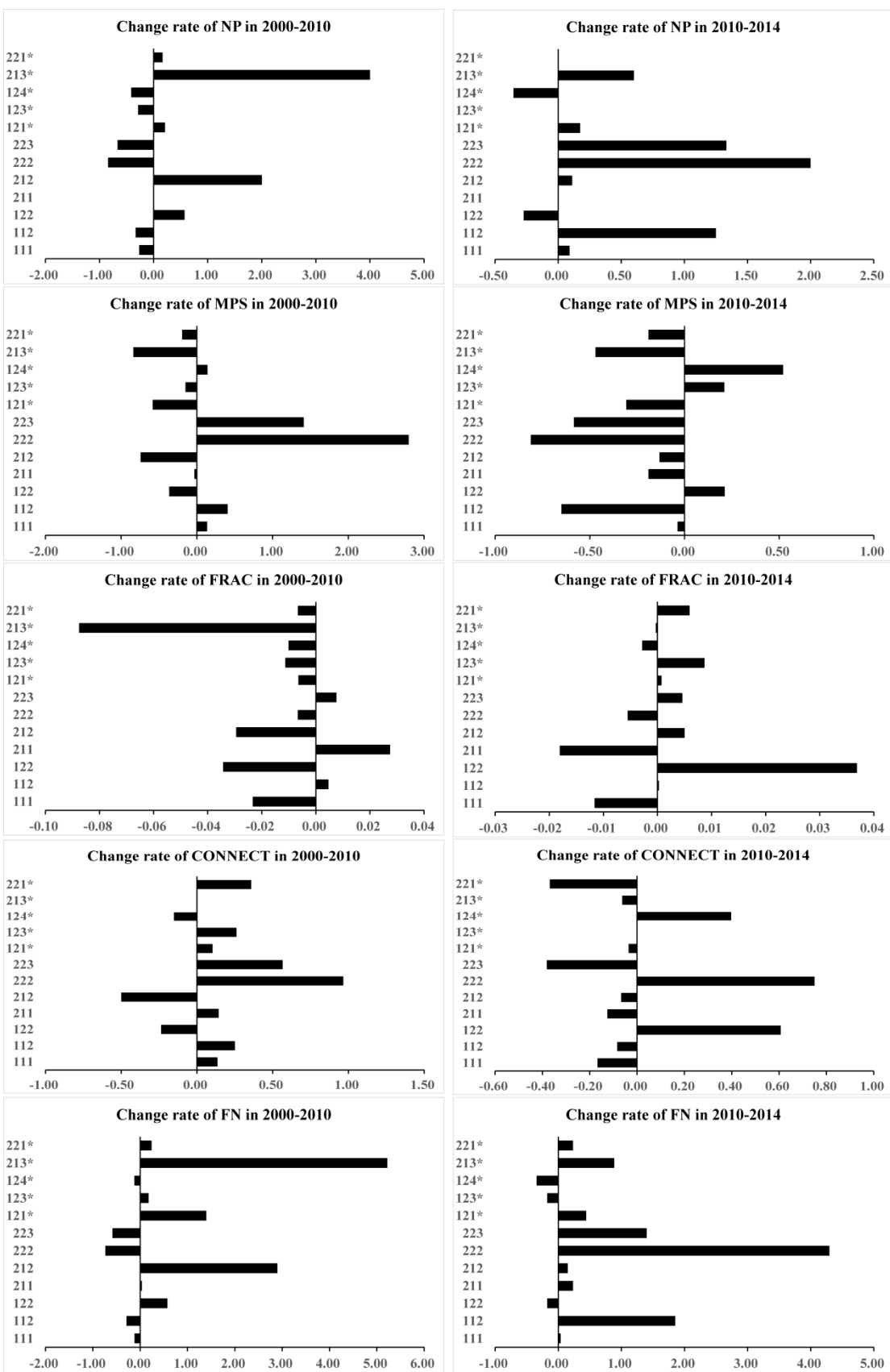

**Figure 7.** Change rate of agricultural land landscape indices for each community in 2000–2010 and 2010–2014. * indicates communities that implemented RFFP. Refer to Table 3 for the abbreviations for landscape metrics.

**Table 8.** Comparison of agricultural land landscape indices between RFFP and non-RFFP communities.

| | RFFP | | Non-RFFP | | | |
|---|---|---|---|---|---|---|
| | **Mean** | **Standard Deviation** | **Mean** | **Standard Deviation** | *t* | *p* |
| **2000** | | | | | | |
| NP | 11.400 | 12.116 | 9.857 | 5.113 | −0.305 | 0.766 |
| MPS | 21.307 | 23.901 | 8.709 | 3.641 | −1.399 | 0.192 |
| FN | 0.163 | 0.196 | 0.136 | 0.067 | −0.343 | 0.739 |
| FRAC | 1.081 | 0.051 | 1.068 | 0.007 | −0.717 | 0.490 |
| CONNECT | 41.382 | 41.903 | 48.764 | 15.663 | 0.433 | 0.675 |
| **2010** | | | | | | |
| NP | 7.800 | 5.586 | 9.143 | 6.890 | 0.358 | 0.728 |
| MPS | 8.468 | 2.519 | 8.895 | 5.214 | 0.168 | 0.870 |
| FN | 0.126 | 0.034 | 0.151 | 0.079 | 0.654 | 0.528 |
| FRAC | 1.057 | 0.023 | 1.057 | 0.010 | 0.006 | 0.995 |
| CONNECT | 52.049 | 28.070 | 60.236 | 27.787 | 0.501 | 0.627 |
| **2014** | | | | | | |
| NP | 8.600 | 2.302 | 10.714 | 7.499 | 0.603 | 0.560 |
| MPS | 6.293 | 3.400 | 5.972 | 2.216 | −0.199 | 0.846 |
| FN | 0.253 | 0.246 | 0.191 | 0.078 | −0.636 | 0.539 |
| FRAC | 1.052 | 0.019 | 1.064 | 0.016 | 1.121 | 0.289 |
| CONNECT | 54.727 | 20.548 | 51.632 | 16.196 | −0.293 | 0.776 |

Refer to Table 3 for the abbreviations for landscape metrics.

### 3.3. Driving Force Analysis of Landscape Change

The degree of fit ($R^2$) and the significance tests of the models showed that the explanatory power of environmental variables on the landscape pattern was greater than 30%, with the probability of F being less than 0.05 (Table S5), indicating that the models were valid. The environmental variables (Table S6) that entered the model through the significance test are shown in Table 9. A significant negative relation was observed between the change in forest fragmentation and the nearness to a township ($\beta = -0.56$, $p = 0.05$), and a significant positive relation was observed between the change in shrubland fragmentation and the proportion of households with solar water heaters ($\beta = 0.64$, $p = 0.02$). A significant positive relation was also observed between the change in agricultural land fragmentation and the mean elevation ($\beta = 0.61$, $p = 0.03$). The entire landscape pattern was significantly negatively related to nearness to a township ($\beta = -0.82$, $p = 0.00$) and the mean elevation ($\beta = -0.56$, $p = 0.03$).

**Table 9.** Stepwise regression analyses: change rate in class/landscape fragmentation regressed on socioeconomic–ecological variables from 2010 to 2014, $N = 12$.

| Variables | Model_Landscape | | Model_Forest | | Model_Shrubland | | Model_Agricultural Land | |
|---|---|---|---|---|---|---|---|---|
| | *p* | *β* | *p* | *β* | *p* | *β* | *p* | *β* |
| Mean elevation | 0.03 | −0.56 | | | | | 0.03 | 0.61 |
| Nearness to township | 0.00 | −0.82 | 0.05 | −0.56 | | | | |
| Solar water heater proportion | | | | | 0.02 | 0.64 | | |
| Intercept | 0.64 | 0.00 | 0.00 | 0.18 | 0.58 | 0.04 | −0.56 | 0.01 |
| Degrees of Freedom | 2,9 | | 1,10 | | 1,10 | | 1,10 | |
| F | 7.93 | | 4.63 | | 7.06 | | 5.92 | |
| *p*-value | 0.01 | | 0.05 | | 0.02 | | 0.03 | |
| $R^2$ | 0.64 | | 0.32 | | 0.41 | | 0.37 | |

The selected variables entered the model through a test of significance (criteria: probability-of-F-to-enter ≤0.05, probability-of-F-to-remove ≥0.10). *β* represents the normalized regression coefficient.

## 4. Discussion

The differences in landscape pattern and fragmentation between RFFP and non-RFFP communities during 2000–2014 were quantitatively compared and validated using of the NP, MPS, FRAC, CONNECT, LFI, and SHDI indices at the landscape level and NP, MPS, FRAC, CONNECT, and FN indices at the class-level. To further examine the driving forces affecting changes in landscape fragmentation in the study area, these indices were regressed on variables representing environmental and socioeconomic factors. In 2000, 2010, and 2014, no significant differences were observed between RFFP and non-RFFP communities in forest, shrubland, agricultural land, and the entire landscape pattern metrics or in their fragmentation. Moreover, no significant differences ($p > 0.05$) were observed in the change rate of each landscape/class index between RFFP and non-RFFP communities from 2000 to 2010, 2010 to 2014, and 2000 to 2014. The primary observations were the reduction in agricultural land area that became more fragmented and the restoration of vegetation that resulted in the benign transformation of the landscape pattern. From 2000 to 2010, the landscape patterns and their dynamic changes between RFFP and non-RFFP communities were variable, with large individual differences among the communities. The RFFP had no significant influence on forests, shrubland, agricultural land, or the entire landscape. After 2010, the fragmentation of the entire landscape increased in both types of communities, with a decrease in forest fragmentation and an increase in agricultural land and shrubland fragmentation. Thus, the RFFP was still not the primary and direct reason for the dynamic changes in forests, shrubland, agricultural land, or the entire landscape pattern in recent years. Many studies suggest vegetation has been restored and forest connectivity and integrity has improved because of the high proportion of agricultural land and bare land converted into forest, shrubland, and grassland. Although forest fragmentation has been effectively restrained, the cover of agricultural land has been greatly reduced and its fragmentation has increased; as a result, the entire landscape pattern is more fragmented [34,37,76]. These results are consistent with those of our study. However, the previous research was based on longitudinal analysis from a time series and focused on the changes in land use and landscape pattern in a certain region before and after the implementation of the RFFP. In the absence of a horizontal comparison, all the changes were attributed to the RFFP, and a pseudo-relationship was established between the RFFP and changes in land use and landscape pattern. Those studies ignored the fact that there might not be an inevitable connection between the changes in landscape pattern and the RFFP [77].

Landscape structure and composition may change dramatically over time in a variety of landscapes [40]. In sustainable landscape development, humans alter the landscape to improve its functionality and create additional value [78,79]. When spatial planning policy is decentralized, local actors should collaborate to decide on the appropriate changes for the landscape to better accommodate their perceived values [80,81]. Facilitating afforestation requires the coordination of land users, officials, and bureaucrats who will have varying concerns and interests in terms of identifying land to be afforested, assemble seedlings, and plant and tend trees. Conflicts and contradictions can arise regarding the means for the improvement of rural livelihoods and the protection and restoration of ecological systems [7,8]. On this basis, the RFFP may not promote the benign transformation of landscape patterns. Meanwhile, rural communities are undergoing rapid social, political, economic and cultural transitions, which directly and indirectly influence the way society interacts with the environment, which in turn can cause rapid change in rural landscape [82]. Hence, landscape pattern changes are strongly affected by the economic, sociocultural and ecological values demanded by its users.

The results of this study showed no significant differences in forests, shrubland, agricultural land, and the overall landscape pattern between RFFP and non-RFFP communities. The dynamic changes were also the same in the two types of communities. These results indicated that the RFFP may not be the most direct cause of changes in the landscape pattern in the study area at the community level; however, the RFFP may act on the landscape pattern together with other natural and economic factors [40,47]. To protect and improve the environmental quality in Weixi County, the RFFP and the

NFPP have been implemented since 2002. In 2008, to increase the income of households and develop the local economy, the government started to encourage the planting of tree crops—particularly walnuts. In 2010, the policy of "linking villages with roads" was implemented, and rural roads were built to connect all villages with asphalt or cement roads. These multiple measures and policies may cause changes in land use and landscape patterns that are interwoven. Therefore, an in-depth analysis should be conducted of the primary driving forces for dynamic changes in the landscape pattern according to the actual local situation to adapt measures to local conditions [45]. Data from interviews and focus group discussions show that at the same time as RFFP was being used to encourage farmland retirement, households in both participating and non-participating communities took marginal, high-elevation farmland out of production. While households in communities that implemented RFFP received subsidies to retire farmland and plant trees, initial tree-planting efforts had limited success. As a result, the program's impact on landscapes was obscured by other factors. In 2008, the government began to encourage walnut cultivation, initially in low-elevation riverside communities in which the climate, soil, and other natural conditions are suitable for walnut growth. By 2010, the "linking villages with roads" program was implemented. Roads brought connectivity between communities and markets that facilitated walnut production [38,43]. Other changes in livelihoods during this period include the adoption of cash crops such as runner beans, maca, and costus root, as well as increasing off-farm labor allocation, varying in magnitude across communities. Under the combined action of government policies, market shifts, infrastructure construction, urbanization, and many other factors, some of the agricultural land in the study region was transformed into vegetation, increasing vegetation coverage, although much of the increase was due to the economic forests. Altogether, forest fragmentation decreased in both RFFP and non-RFFP communities during 2000–2014, with the decrease particularly pronounced after 2010, as the gradual effects of farmland retirement generating vegetation through tree cultivation or forest succession became evident. Thus, vegetation cover increased and forest landscape patterns were consolidated in the study area, which is consistent with government objectives [9], but the fragmentation of the overall landscape pattern increased [35].

Fragmentation and landscape metrics at the landscape level were not significantly different between RFFP and non-RFFP communities, indicating that the RFFP did not cause differences in the landscape pattern at the community level. Looking at change over time, from 2000 to 2014, landscape fragmentation increased in communities with and without the RFFP, possibly due to the implementation of the "linking villages with roads" policy in this region. The construction of roads and the development of some economic activities often causes the fragmentation of vegetation, rivers, and other landscape features [83,84]. Regression results showed that the nearness to a township and elevation—not the RFFP—were the main environmental factors that affected change in overall landscape patterns. Communities that are closer to towns and lower in elevation have higher landscape fragmentation than those farther away at higher elevation, which may be due to more frequent infrastructure construction and economic activity that occurs in these communities [85]. As a result, the landscape segmentation and the number of patches increased, the mean patch area and the connectivity decreased, and the landscape fragmentation increased. The nearness to a township was also an important driving force for the change in forest fragmentation patterns, although the contribution of the RFFP was again not significant. In communities far from a township, because of the lack of transportation and over-reliance on the cost of selling crops to make ends meet, households preferred to choose long-term work away from home. The choice to leave the area for work reduces population pressure and external interference and promotes a reduction in forest fragmentation [86] (Figure 8). The proportion of households with solar water heaters was the main driving force affecting the change in shrubland landscape pattern, with the effect of the RFFP also not significant. The households of Weixi County typically rely on trees for fuelwood, and the use of solar water heaters can reduce pressure on forests [87]. Households can also use solar water heaters to heat livestock fodder. Although farmers may spend less time and energy harvesting fuelwood, they may meet residual fuel needs by obtaining dead branches from shrubland. The influence of elevation on agricultural land fragmentation was

significant. The agricultural land fragmentation increased for both RFFP and non-RFFP communities, which indicated that the implementation of the RFFP was not the key factor in the decision to change land use, because the communities without the RFFP also redistributed a large amount of agricultural land [18,38]. The ratio of input to output may be the primary consideration when farmers convert agricultural land into other land use types [16,18,46]. Because of the cold climate, low soil fertility, and high cost of crop transportation in high-elevation communities, households may conclude that the returns do not justify investing resources in crop cultivation. Thus, high-elevation farmland is likely to be abandoned or converted to other uses. Ding, et al. [88] analyzed the factors that influence land use and landscape pattern changes in hilly regions and concluded that the changes were jointly affected by hydrology, climate, topography, soil, vegetation, and elevation. The current study shows that rural landscape fragmentation was influenced not only by the natural environment but also by policy programs, socioeconomic conditions, and the livelihoods of households [30]. Based on these results, the government should take all these factors into account when formulating relevant policies [89,90].

This study linked social, economic, and ecological data with remote sensing data across time at a fine spatial scale. This approach places some constraints on the analysis. The sample size selected was 12 communities in one county. This limited sample does not allow inference regarding the situation with the RFFP in other regions. Differing resolutions of remote sensing images can also affect the interpretation of results of changes in land use, although this error was minimized by resampling and other measures. The questionnaire survey and focus groups both involved a discussion of the use of forests by households, and while we took efforts to minimize recall and contextual biases, they cannot be completely eliminated. Finally, because this research focused on the community level, there may be risks associated with extending the results to larger scales.

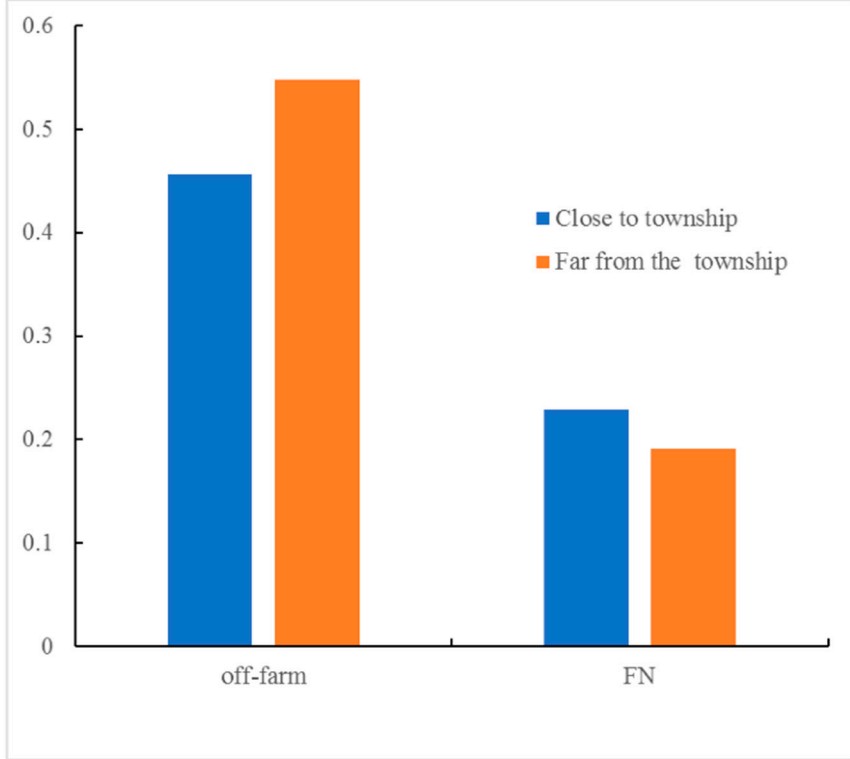

**Figure 8.** Effects of distance from a township on off-farm labor and forest fragmentation. Off-farm, the mean value of proportion of households with off-farm labor in 2013; FN, the mean value of forest fragmentation in 2014.

## 5. Conclusions

The hypotheses in this study that forest and shrubland fragmentation would decrease and farmland fragmentation would increase in the RFFP communities and that forest and shrubland fragmentation would increase and farmland fragmentation would decrease in the non-RFFP communities were not corroborated. No significant differences were observed in forests, shrubland, agricultural land, and the entire landscape pattern or in fragmentation between the communities with and without the RFFP in 2000, 2010, and 2014. Similarly, the dynamic changes observed during 2000–2010 and 2010–2014 were the same between RFFP and non-RFFP communities. This result suggests that the RFFP was not a direct cause of the changes in landscape patterns and fragmentation, and therefore, the evaluation of the RFFP should be combined with other processes that affect land cover change. The change in landscape patterns in the study area was influenced not only by natural conditions but also by other factors such as economic change, household livelihoods, and policy interventions. When the implementation of national policies and other natural and socioeconomic activities affecting land use interact, the ways these various processes intersect must be carefully evaluated [26,91,92].

Our results provide an empirical basis for the exploration of the evolution of landscape patterns and the formulation and implementation of the RFFP. China has started a "second round" of the RFFP and has used satellites to monitor changes in land cover and vegetation [2]; however, policy implementation is likely to be hampered without accounting for socioeconomic variables and participant livelihoods. Vegetation can regulate the climate and provide habitats and other services for life, depending on the management and the environmental conditions [93]. Considering rural people's livelihoods and market dynamics, the cultivation of tree crops may be an important direction to follow in future RFFP policy. The success or failure of the next round of the RFFP may depend on the extent to which the formulation of the relevant policies accounts for factors affecting participation and consolidation. This study only investigated 12 communities in Weixi County, which is a limited sample, and therefore, the specific effects we find may only reflect conditions that prevail in this region of southwest China. The determination of how the RFFP and other related processes play their roles in other regions should be adapted to the local conditions.

**Supplementary Materials:** The following are available online at http://www.mdpi.com/1999-4907/10/10/933/s1, Table S1: Classification accuracy (%) and Kappa coefficient (%) of the Lancang River basin in Weixi County in 2000, 2010, and 2014; Table S2: Comparison of dynamic changes in landscape indices between RFFP and non-RFFP communities, Refer to Table 3 for the abbreviations for landscape metrics; Table S3: Comparison of dynamic changes in forest landscape indices between the RFFP and non-RFFP communities, Refer to Table 3 for the abbreviations for landscape metrics; Table S4: Comparison of dynamic changes in shrubland landscape indices between the RFFP and non-RFFP communities, Refer to Table 3 for the abbreviations for landscape metrics; Table S5: Comparison of dynamic changes in agricultural land landscape indices between the RFFP and non-RFFP communities, Refer to Table 3 for the abbreviations for landscape metrics; Table S6: The variables adopted in the study and the corresponding methods.

**Author Contributions:** Conceptualization, W.L., J.A.Z. and Z.Z.; Data curation, W.L. and Z.Z.; Formal analysis, W.L., J.A.Z. and Z.Z.; Funding acquisition, J.A.Z. and Z.Z.; Investigation, W.L., J.A.Z. and Z.Z.; Methodology, W.L. and Z.Z.; Project administration, J.A.Z. and Z.Z.; Resources, W.L., J.A.Z. and Z.Z.; Software, W.L.; Supervision, Z.Z.; Validation, W.L. and Z.Z.; Visualization, W.L.; Writing—original draft, W.L. and Z.Z.; Writing—review & editing, W.L., J.A.Z. and Z.Z.

**Funding:** This study was supported by grants from the National Natural Science Foundation of China (41761040 and 41361046) and the National Science Foundation, award number SMA-1415028.

**Conflicts of Interest:** The authors declare that they have no competing interests.

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
