# Peer review of "Does the “Returning Farmland to Forest Program” Drive Community-Level Changes in Landscape Patterns in China?"

_forests, doi:10.3390/f10100933_

Round 1
Reviewer 1 Report
This paper examines the landscape effects of China’s Returning Farmland to Forest program (RFFP) at the community level. A mixed methods approach is taken – using spatial-temporal analysis of remotely sensed data as well as household interview, focus group, and survey data. The approach is not particularly novel, but the empirical aspects of the study appear well executed. There are several areas in which the manuscript should be improved before publication.
(1) The introduction is heavy on specific details of prior work with RFFP and waits too long to introduce the main objectives of the study at hand. Further, this section is in need of broader literature review including theory and other empirical work surrounding similar research but outside of RFFP.
(2) Related to the needed improvements in the introduction, the authors need to better connect their findings to theory. Although the authors are convincing in their empirical evidence that RFFP is ineffective, they fail to frame this insight in terms of its theoretical significance – and also fail to use theory to help us understand why other factors may be driving landscape change.
(3) Although the procedural aspects of the methods are clear, the analytical aspects are not. This is of lesser concern, but specifically, it is difficult to understand how the survey/focus group/and interview data was analyzed and combined with the spatial analysis. This could use some (though not substantial) additional clarity. It is difficult to discern which parameter was derived from which method – perhaps a table could clarify this. Methods are also interspersed throughout the results section and need to be moved back into the methods section. I have noted these in comments on the attached PDF.
(4) Tables and figures could use some improvement for clarity – also, I’m not certain it is necessary to include tables with non-significant results in the main text. This could be moved to supplementary material.
Please also see attached PDF. Minor errors are highlighted and there are a few comments for the authors.

Author Response
Point 1: The introduction is heavy on specific details of prior work with RFFP and waits too long to introduce the main objectives of the study at hand. Further, this section is in need of broader literature review including theory and other empirical work surrounding similar research but outside of RFFP.
Response 1: Thank you for your constructive comments, which are of great value for improving our manuscript. We strongly agree with your opinion that the lead-in before the introducing the main objectives is really too long. We have reduced this section, while extending the literature review to include theory and other empirical but similar research beyond that of RFFP. Please refer to lines 43-150 of the Introduction for the revised text.
Point 2: The research questions or specific objectives need to be more clearly stated earlier on in the introduction.
Response 2: Yes, this caveat is a good suggestion, one that we strongly agree with. We have clearly stated the research questions in the second paragraph of the Introduction. See lines 97-98 of the Introduction for the revised text.
Point 3: What does "elastic change" mean?
Response 3: It refers to the ability of the landscape pattern to resist interference[1,2]. In order to make the introductory text more concise, the first paragraph in which "elastic change" appeared has been deleted.
Point 4: What is the "study period" - please state/restate
Response 4: "Study period" in that sentence refers to the years 2000, 2010, 2014. The corresponding revised text can be found on line 150 of the Introduction.
Point 5: What is a "natural village"?
Response 5: "Natural village" is a village formed naturally by villagers who live together in a natural environment for a long time. It is a basic administrative unit that makes up an administrative village. Administrative village is a village-level administrative unit managed by the township government and generally is composed of one or several natural villages. We selected 12 communities across four administrative villages in two townships[3], of which five took part in the RFFP.
Point 6: Related to the needed improvements in the introduction, the authors need to better connect their findings to theory. Although the authors are convincing in their empirical evidence that RFFP is ineffective, they fail to frame this insight in terms of its theoretical significance-and also fail to use theory to help us understand why other factors may be driving landscape change.
Response 6: Thank you for your feedback. This is unfortunate. We have added commentary on the theoretical significance content to the Discussion section. Please see lines 467-480 of the Discussion subsection 4.1 for the revised text. If you have a better suggestion, we’d be willing to try it. Hopefully, this is now better understood.
Point 7: Although the procedural aspects of the methods are clear, the analytical aspects are not. This is of lesser concern, but specifically, it is difficult to understand how the survey/focus group/and interview data was analyzed and combined with the spatial analysis. This could use some (though not substantial) additional clarity. It is difficult to discern which parameter was derived from which method – perhaps a table could clarify this. Methods are also interspersed throughout the results section and need to be moved back into the methods section. I have noted these in comments on the attached PDF.
Response 7: Yes, we have used a table to clearly clarify the analysis aspects. Survey data were aggregated into community-level measures of crop cultivation, livestock husbandry, RFFP participation, and non-farm labor allocation. Landscape structure and fragmentation data from the remote sensing analysis were incorporated into the dataset. We tested relationships between landscape fragmentation and biophysical, policy, and socioeconomic predictors using bivariate comparisons and ordinary least squares regression models. Methods that were interspersed throughout the results section have been moved back into the methods section (see lines 286-299). The parameters investigated together with the methods employed are as follows:
Table S6 The variables used in the study and the corresponding methods
Parameters |
Method |
Change in forest fragmentation in 2010–2014 |
Interpretation of remote sensing data and calculation of landscape indices |
Change in shrubland fragmentation in 2010–2014 |
Interpretation of remote sensing data and calculation of landscape indices |
Change in farmland fragmentation in 2010–2014 |
Interpretation of remote sensing data and calculation of landscape indices |
Change in landscape fragmentation in 2010–2014 |
Interpretation of remote sensing data and calculation of landscape indices |
Mean elevation |
Remote sensing data |
Household density |
Interview and survey data and remote sensing data |
RFFP implementation |
Interview and survey data |
Off-farm proportion |
Interview and survey data |
Nearness to township |
Interview and survey data and remote sensing data |
Solar water heater proportion |
Interview and survey data |
Walnut area |
Interview and survey data |
Cropland retirement proportion |
Interview and survey data |
Planted trees proportion |
Interview and survey data |
Mean patch area (MPS) |
Interpretation of remote sensing data and calculation of landscape indices |
Number of patches (NP) |
Interpretation of remote sensing data and calculation of landscape indices |
Fractal dimension index (FRAC) |
Interpretation of remote sensing data and calculation of landscape indices |
Connectance (CONNECT) |
Interpretation of remote sensing data and calculation of landscape indices |
Fragmentation index (class- or landscape-level) (FN/LFI) |
Interpretation of remote sensing data and calculation of landscape indices |
Shannon's diversity index(SHDI) |
Interpretation of remote sensing data and calculation of landscape indices |
Point 8: Tables and figures could use some improvement for clarity – also, I’m not certain it is necessary to include tables with non-significant results in the main text. This could be moved to supplementary material.
Response 8: Thank you for your suggestion. The tables with non-significant results in the main text analyze the effectiveness of RFFP from a cross-sectional perspective and the corresponding figures analyze the effectiveness of RFFP from a temporal perspective. So, it is also important to keep tables with non-significant results in the main text. In terms of the figures, since the spatial distribution of the changing areas in figure 2 is scattered, if the changing areas are highlighted, and the land use types cannot be clearly displayed. In order to clearly show changes in land cover and landscape pattern, it may be better to keep the original figure 2. If you have a better suggestion, we’d be willing to try it.
Point 9: "empirical" instead of "theoretical"?
Response 9: Thank you very much for correcting our text. See line 582 of the Conclusions for the revised text.
Point 10: The presentation of this graph (Figure 8) is slightly confusing?
Response 10: Figure 8 shows the effects of distance from township on off-farm labor and forest fragmentation. What we want to convey through Figure 8 is the distance from the township may affect forest fragmentation by affecting the proportion of off-farm labor. In communities far from a township, because of the lack of transportation and too much reliance on the cost of selling crops to make ends meet, households preferred to choose long-term work away from home. The choice to leave the area for work reduces population pressure and external interference and promotes a reduction in forest fragmentation.
References
Turner, M.G.; Romme, W.H. Consequences of spatial heterogeneity for ecosystem services in changing forest landscapes: Priorities for future research. Landscape Ecology 2013, 28, 1081-1097. Wiens, J.A. Is landscape sustainability a useful concept in a changing world? Landscape Ecology 2013, 28, 1047-1052. Zinda, J.A.; Zhang, Z. Explaining heterogeneous afforestation outcomes: How community officials and households mediate tree cover change in china. World Development 2019, 122, 385-398.

Reviewer 2 Report
English language and style require moderate improvements.
The introduction addresses the main issues but it may be reduced as it is too long and the results from previous studies may be reduced.
The document contains several grammatical errors, for example:
Page 1, Line 41: Change "throngh" with "through"
Author Response
Point 1: The introduction addresses the main issues but it may be reduced as it is too long and the results from previous studies may be reduced.
Response 1: Yes, thank you very much for your valuable and constructive comments. We have reduced the text on results from previous studies. Meanwhile, the logical structure of the introduction has been improved to make the text more concise. The corresponding revised text can be found on lines 43-133 of the Introduction.
Point 2: English language and style require moderate improvements.
Response 2: Yes, we have asked a native English speaker to help us via a professional English editing service. John Aloysius Zinda, co-author of this article, has also improved the English language and style of the article.For details, please see the attached editorial certificate in the word document.
Point 3: The document contains several grammatical errors, for example: Page 1, Line 41: Change "throngh" with "through"
Response 3: Yes, thank you for spotting this typo. We went over the article again and these several grammatical errors have been corrected.
